# Reform of the Health Insurance Funding Model to Improve the Care of Noncommunicable Diseases Patients in Saudi Arabia

**DOI:** 10.3390/healthcare10112294

**Published:** 2022-11-16

**Authors:** Ahmed Hazazi, Andrew Wilson, Shaun Larkin

**Affiliations:** 1Menzies Centre for Health Policy and Economics, Sydney School of Public Health, University of Sydney, Sydney, NSW 2006, Australia; 2Public Health, Faculty of Health Sciences, Saudi Electronic University, Riyadh 13316, Saudi Arabia

**Keywords:** reform, health insurance, health systems financing, non-communicable diseases care, Saudi Arabia

## Abstract

Health insurance models are being considered as part of health system reforms in Saudi Arabia. This paper assesses the attributes of health funding models that support better control of non-communicable diseases (NCDs) and perspectives on health insurance as a model from the perspective of patients, clinicians, and managers. The study employed a mixed-methods research design that included quantitative and qualitative data gathering and analysis. Study findings indicated concerns that the current health funding mechanism is financially unsustainable and, as a result, there will be a greater reliance on personal health insurance to support government spending on healthcare. Essential elements of any health insurance model to support effective NCD management identified from a review of the literature and interviews include the following: ensuring continuity of care and equity; funding chronic disease prevention interventions; prioritising primary healthcare; and maintaining the principle of community rating to prevent insurers from discriminating against members. Other desirable attributes for the funding model includes collaboration across primary, secondary, and tertiary care. Healthcare finance reform aimed at adopting and increasing personal health insurance coverage may play a critical role in extending access to healthcare, eliminating health inequities, enhancing population health, and reducing government spending on healthcare if appropriately considered.

## 1. Introduction

The public healthcare system in Saudi Arabia is overseen and financed by the government through the Ministry of Health (MOH) and care is provided free to all Saudi citizens in accordance with the country’s constitution [1]. The MOH also manages and supervises private sector services and other government healthcare sectors to achieve the government’s goals and implement its strategies. The MOH provides and funds 57% of healthcare services through 287 hospitals and through a network of 2361 primary healthcare centres distributed around the country [2]. Other quasi-governmental agencies provide a further 10% of healthcare services via 50 hospitals servicing employees and their family members. The private sector provides 33% of services through 167 hospitals [2].

The Saudi government has introduced reforms to improve health system financing and delivery in the country. The reforms highlight several challenges in meeting the healthcare needs and the rising cost of healthcare. Healthcare cost increases result from population growth and ageing, the increase in non-communicable chronic diseases (NCDs), and the cost of providing modern healthcare (more and more costly diagnostic and therapeutic options). Given the Saudi government funds 67% of healthcare, this is raising concerns about the financial sustainability of the current funding model.

Revenue from oil exports, which accounts for over 90% of government revenue [3,4], has been the primary source of public sector funding. The uncertainty about oil prices in international markets, and its long-term demand, is a cause of concern about the sustainability of public finances, and healthcare is an increasing proportion of public finance expenditure. Private health insurance is seen as a way of reducing government healthcare expenditure. The establishment of the Council of Cooperative Health Insurance (CCHI) in 1999 was an important milestone in the move to private health insurance [5]. The Cooperative Health Insurance (CHI) program was to be introduced to achieve this over three phases [6,7]. In the first phase, the CHI has been applied to Saudis and expatriates employed in the private sector, where employers cover employee health insurance expenses. In the second phase, the CHI would be applied to citizens and non-citizens employed by the government in the public sector, paid by the government. The CHI would then be applied to other groups, such as visitors, in the third phase of development. However, the final decision on the programme’s second and third phases are yet to be made. Researchers and policymakers see the health insurance programme as vital for improving the accessibility to and financial sustainability of the Saudi healthcare system [8,9,10,11].

In 2016, Saudi Arabia announced the “Vision 2030” strategy, an ambitious development plan with the primary objective of reforming the country’s economy by 2030 [12]. One of the objectives for health is reforming the health financing system to confront the current system’s challenges with further endorsement of the move toward private health insurance. A stated objective of this is that all covered individuals can access the necessary healthcare services at the time of need without extra financial burdens [13]. Currently, the MOH is considering applying new reforms with a particular emphasis on improving the finance system and institutionalising the health system to ensure service delivery competency and success in line with Saudi Vision 2030 [12].

The way a healthcare system is funded has a significant influence on the way healthcare is provided and whether the organisation and provision of care aligns with what is known about effective care for patient groups, such as people with NCDs. Health insurance, a key component of Saudi health funding, has been recognised as having the potential to impact on NCDs management in primary healthcare settings [14,15,16].

This paper aims to assess the desired attributes for any health financing model to improve the control and management of NCDs. It assesses the perspective of patients with NCDs, clinicians, and policymakers on the reform of the Saudi health insurance funding model. Given that the planning and decision making for the Saudi funding model is ongoing, this study aims to inform decisions about the design of any health insurance model to improve and not be a barrier to a more effective and efficient health system approach to NCDs.

## 2. Materials and Methods

This research employs a mixed-methods research design that included quantitative and qualitative data gathering and analysis [17]. Using semi-structured interviews this study explores the views of healthcare professionals and managers about NCD control and the proposed health financing model. Using a self-administered questionnaire, it also examines the perspective of NCDs patients in Saudi Arabia on NCD care arrangement and healthcare costs. These were combined with the results of a review of literature to identify issues and desired attributes of health financing models relevant to improving NCD management.

Qualitative semi-structured interviews were undertaken to gather information about policies and strategies in relation to health insurance with stakeholders. Interviewees were selected to contribute based on their expertise by using purposive and snowballing methods [18]. A sample of 25 health providers and managers from the MOH were interviewed. Recruitment was ceased after 25 interviews as it was evident that no new themes were emerging (saturation) [19]. The interviews were audio-recorded, transcribed, and coded into key themes. Interviewees were asked to recommend or provide any relevant policy statements or documents. A thematic analysis framework was used for the data analysis consisting of the following six steps: becoming familiar with the data; searching for themes; reviewing themes; de-fining and naming themes; and writing the research report [20]. The NVivo software was used to organise and code the data [21].

The cross-sectional patient survey took place between May 2019 and July 2019, with the questionnaire completed anonymously by 315 patients with NCDs attending (MOH) Primary Health Care Centres (PHCs). The patients were recruited from 20 of 58 Ministry of Health (MOH) PHCs in urban Riyadh, Saudi Arabia. The patient assessment of chronic illness care (PACIC) was used to assess the quality of care received by patient with NCDs. Here, PACIC was selected for this study because it allows for a standardised assessment of the patient’s perspective on the current quality of healthcare services, which is essential in the evaluation of chronic care management [22,23]. The PACIC questionnaire has been validated and used in the Saudi context [24]. For an estimated patient population of 20,000, the sample size for estimates of PACIC scores with a 5% margin of error at 95% confidence level, was 377 (calculated using a Raosoft calculator) [25], which was simplified as a target of 400. The sample size achieved was 315 due to incomplete questionnaires giving an effective response rate of 79%. This sample size is similar to that used to develop and validate the PACIC instrument [26].

Only patients with NCDs were recruited for inclusion in the study. The inclusion criteria were patients with at least one chronic disease who had attended a NCD clinic at a PHC during the previous six months. All patients meeting these criteria and attending a clinic on study days were approached to participate in the study. The patients were provided with information about the study and asked to provide written consent. Anonymous self-report questionnaires were used to collect the data, which were analysed using the Statistical Package for the Social Sciences (SPSS) Version 26.0. Descriptive statistics, including frequencies and percentages, are reported, and logistic regression analyses were performed to identify the demographic factors associated with patients’ perceptions of chronic disease management. Ethical approval was given by the ethics committee of the MOH (IRB log no: 2019-0028 E) before data collection commenced.

## 3. Results

The final patient survey sample consisted of 315 patients with NCDs. The descriptive statistics and sociodemographic and medical characteristics of the sample were published elsewhere [27]. In brief, the mean age of the patients was 56 years, ranging from 29 to 85; 49.8% were male and 50.2% were female, with more than half (52.4%) belonging to the 50 to 64 age group. Graduate level education was reported by over a third (37.1%), and 44.4% reported having been diagnosed with an NCD for less than 5 years. Among the study sample, 75.6% reported having only 1 NCD, while 24.4% had more than 1 NCD. Diabetes was the most common NCD, reported by 43% of the study population, followed by hypertension in 21.4% patients. Almost 50% of the diabetes patients also had hypertension. The presence of NCDs among the patients is given in Table 1. Binary logistic regression analysis was performed for the association between sociodemographic and disease related variables with selected aspects of patient experience. The statistical significance of the results was accepted with a *p*-value ≤ 0.05, as shown in Table 2. From the binary logistic regression analysis controlling for age, gender, educational attainment, and income, patients reported they were happy with the possibility of private health insurance as a means to access any hospital (*p* < 0.01). Patients who had NCDs a for shorter period (less than 20 years) are more likely to prefer private health insurance to access healthcare services (*p* < 0.02) (*p* < 0.02) (*p* < 0.01). Patients who work in the government sector showed a strong preference and acceptance for private health insurance (*p* ≤ 0.01). Survey respondents prioritised improvements in organised care that is comprehensive and focused on their needs and that helps them to identify clear goals for their treatments and become more involved in the management of their illnesses [27].

Themes from the analysis of the interviews with physicians and health system managers are grouped according to whether they refer to new reforms (improving health financing), private health insurance, strengths, weaknesses, and challenges.

New reform (improving health finance).

Interviewees reported that, in recent years, the MOH has been seeking alternative funding sources that contribute to reducing public health service expenditure. The goal of this reform is to improve the funding mechanism and to improve the healthcare services offered by PHCs and hospitals. Interviewees stated that they believed that the Saudi government had the political will to reform the healthcare finance system, improve healthcare facilities, provide quality healthcare services, and reduce healthcare costs. As the cost of healthcare services is constantly rising, interviewees believe that health insurance, which would see greater contributions from individuals or third parties, such as employers, is a strategic solution to strengthening the healthcare financing system. One interviewee stated that, “The health sector is facing a huge inflation in demand for health care and experiencing constant growth in spending on it … and in response to this growing demand and huge expenditures, it is necessary to look for new ways to finance and operate medical institutions” (Interviewee no. 7, an executive director). Furthermore, interviewees argued the importance of improving health financing, with one stating “So, it has become necessary to move towards health insurance to sustaining health care services for all members of the community” (Interviewee no. 14, an executive director).

According to the interviewees, Saudi Vision 2030 recognises the importance of health insurance in both economic growth and decreasing government spending on healthcare. Furthermore, Vision 2030 places a lot of emphasis on the private sector; hence, the government is hoping for more private sector investment. Therefore, the MOH has advocated increasing the private sector’s contribution in health spending through alternative financing and delivery mechanisms, specifically health insurance and collaboration with the private sector.

### 3.1. Private Health Insurance

Interviewees reported that an essential goal of the reform is to improve the quality and efficiency of health services. It was reported that the implementation of health insurance would turn the healthcare finance challenges into opportunities to ensure comprehensive healthcare for citizens while making the health sector more efficient and competitive. Furthermore, interviewees reported that the implementation of health insurance is of great importance to the economy. It was reported as contributing to improving healthcare by increasing access to health services for community members, creating an incentive to provide high quality health services. A health insurance model was seen as creating a competitive environment between health service providers, which could positively influence the level of satisfaction of recipients while easing the burden or pressure on government hospitals.

One interviewee stated that, “… the government aims to ensure that the new health insurance programme does more than just cost shifting; it also wants to ensure that all residents have access to high-quality health care” (Interviewee no. 3, an executive director).

Another interviewee emphasised the gaps in current insurance arrangements which focus only on treatment. The gap was described by one interviewee as follows: “the current private health insurance is designed to cover the patients’ treatments only and does not focus on the prevention” (Interviewee no. 21, a physician).

Another interviewee stated the following: “… we want prevention-based insurance that covers all cases” (Interviewee no. 16, a physician). Interviewees’ feedback highlighted that the future health insurance should focuses on health promotion and prevention, to support the management of NCDs.

#### 3.1.1. Strengths of the Current Approach

Interviewees reported that a significant strength of the current health system is that the Saudi government has demonstrated a solid commitment to improving healthcare services and has given high priority to the development of primary, secondary, and tertiary healthcare services. Respondents pointed to the increased focus on primary healthcare centres to enable universal access and investment in improving health records. Likewise, having a national strategy for reforming the health sector and adopting a plan that includes all the health authorities was seen as a strength, as was having clear strategies, such as Vision 2030.

#### 3.1.2. Weaknesses of the Current Approach

Interviewees noted that despite substantial government funding, the healthcare system continued to struggle to meet all the healthcare demands. Organisational weaknesses identified included limited coordination between government sectors and private sectors, which allows duplication of health services and financing for the same beneficiaries. This, for example, allows citizens in larger centres to have access to different government and private providers, with a potential waste of resources and inequity.

One interviewee stated that, “…government or private sectors hospitals throughout the country provide double services for some individuals, which means double costs, whether in human resources or equipment” (Interviewee no. 19, a physician).

Furthermore, the payment models used to reward healthcare professionals and hospitals, namely a salary-based model for the government sector and a fee-for-service (FFS) model for the private sector, were also mentioned by the interviewees as another weakness of the existing healthcare system. This difference in remuneration was a potential driver of inefficient use of resources where individuals could access both public and private services. Accordingly, the interviewees believe that reforming the current payment system and developing a new purpose-build payment model is essential to realise better quality and more efficient services.

### 3.2. Challenges

Interviewees noted that, while the Saudi healthcare system has improved over the last two decades, it still faces several challenges. An increasing burden of chronic diseases, limitations of the electronic health records system, and a lack of cooperation and coordination with other sectors of care were challenges identified to be addressed to meet the growing demand for healthcare services. The growing population, rising costs of healthcare services especially for chronic diseases, inequitable access, and increased expectations for higher quality services were reported as challenges. Concerns were expressed about the sustainability of the current free healthcare services and their ability to meet these growing population demands, particularly given uncertainties of government revenue given its dependency on oil sales.

## 4. Discussion

Saudi Arabia is facing challenges in its primary healthcare system. These challenges include increased demand for healthcare because of rapid population growth, poor cooperation and coordination between sectors of care, inequitable access, and a growing burden of chronic diseases [28,29]. The population of Saudi Arabia is about 34.4 million people, and the country’s life expectancy is anticipated to reach 75.2 years by 2025 [30]. It is anticipated that, in 2035, 44% of the population will be over the age of 40, and 14% will be over the age of 60, creating a need for healthcare to cope with NCDs, which are estimated to account for 78% of all deaths by 2025 [31]. These trends will result in an increased demand for specialist medical and treatment, which will increase pressure on public health facilities and increase the government’s financial burden in relation to healthcare.

Saudi Arabia’s public healthcare system is supervised and funded by the government through the MOH and is offered free of charge to all citizens in accordance with the country’s constitution. Several reforms are being made by the Saudi government to enhance the financing and delivery of this healthcare system. Despite the attempts to improve the healthcare system, there are several challenges in meeting the healthcare needs and the rising healthcare expenditures. The rapid increase in healthcare expenditures associated with population expansion in general, and NCDs in particular, continues to impose enormous upward pressure on the government, putting it at risk of financial instability. As a result, private health insurance was established in Saudi Arabia to assist in funding healthcare and reduce government spending.

### 4.1. Reform

The government has shown a strong commitment to improve individuals’ health by giving high priority on the development of primary, secondary, and tertiary healthcare services and providing free healthcare to Saudi citizens. Like many other countries, Saudi Arabia is challenged with increasing healthcare costs [32,33,34,35]. Approximately 8% of the government budgetary expenditures (SAR 82 billion or US $21.8 billion) were dedicated to the health sector in 2020 [2]. The government believes, as reflected in responses from our interviews with senior health managers, that the current model of financing and delivering healthcare is inefficient and unsustainable, particularly considering uncertainties about the main source of public revenue, oil sales. Other studies in Saudi Arabia have also questioned the future sustainability of the current healthcare financing model [36].

In recent years, the MOH has been suggesting reforms that focus on changes to the health system’s organisation and financing to bring more efficiency and effectiveness in service delivery [12]. During the development of the process for reforms, there was no mention of any metrics that might be used to evaluate the effectiveness of the modifications or assess the overall success of the reforms. Thus, the impact of the reform process in the Saudi health sector is difficult to assess since it is difficult to determine the outcomes of the changes and their influence on the health sector. There has also been no research to assess the impact of these reforms on the Saudi health system. The lack of detail in the announced reform measures also make it difficult to assess the areas in which there may be impacts on people’s health. Previous research suggest that the proposed reforms may not have had much impact since they are not comprehensive [11]. It is a concern that changes to a fundamental aspect of the health system, its financing, even if necessary for the overall economy, could have unintended consequences without consideration of the overall system.

Saudi Arabia’s “Vision 2030” aims to bring about fundamental changes in the healthcare financing and delivery system to improve access to quality healthcare for its population. The government’s “Vision 2030” places a high priority on private sector development and greater private sector investment. The government is looking to private sector investment to advance healthcare access, efficiency, efficacy, and quality of treatment while reducing dependency on oil export income and government expenditure. The plan, which includes various health-related improvements, has a particular emphasis on the private health insurance as the primary funding mechanism in place of funding from government revenues.

### 4.2. Health Insurance

While there is a plan to mandate health insurance, the form of health insurance that would be mandated has not yet been identified. Any reform of health insurance in Saudi Arabia should consider the wide variability and unequal distribution of healthcare costs borne by individuals, especially those with chronic conditions. Most experts consider that, of the health insurance models possible, a broad-based social insurance model with government subsidy would provide the best coverage with the least additional burden on the policyholders [11]. The underlying principle behind social insurance is that it is mandatory for everyone to be insured, albeit that some may require a public subsidy to achieve this, and that essential healthcare benefits are available to everyone. Furthermore, the aims of social health insurance are preventing or reducing large out-of-pocket expenditures, increasing proper utilisation of health services, providing universal healthcare coverage and improving health status [37,38]. Social health insurance models have operated successfully for decades in many countries, most notably in Germany. Social insurance schemes differ significantly from private insurance policies. Private health insurance schemes are generally more variable in enrolment, coverage, and cost, often with an expectation that they will make a profit. At present, the government is implementing a private health insurance model.

Our results showed that patients with NCDs would probably be accepting of a private health insurance model providing they retained the same access to services, but we did not clarify respondents understanding of private health insurance. Our study results are consistent with those in another study that explored individuals’ preferences regarding health insurance schemes and found that individuals in Saudi Arabia are willing to obtain health insurance to access private health facilities for quality healthcare services [39]. Moreover, over the previous two decades, private health insurance in Saudi Arabia has witnessed unprecedented growth, with the number of covered members rising from 3.3 million in 2006 to 9.8 million in 2022. The population covered by private health insurance more than doubled from 13% in 2006 to 28% in 2022 [40]. Wherry and Miller [41] indicated that private health insurance results in increased use of healthcare services and higher rates of chronic disease diagnosis. This suggests that health insurance could lead to an increase of utilization of healthcare services. This could potentially lead to improvements in population health if there is existing underutilization of effective healthcare interventions. However, it could also lead to substantial increases in healthcare costs without improvements in population health if the increased servicing is not appropriate or not filling unmet needs. Countries, such as the United States, with high dependency on private or employer health insurance and fee-for-service medicine, have witnessed some of the highest growth in healthcare costs and massive increases in health insurance premiums despite high levels of government subsidy for some patient sectors, such as the elderly [42,43].

Much is known about what works in healthcare to improve outcomes for NCDs and, given that NCDs cause some of the highest demands on the health system, it is essential that these are supported in any private health insurance model. We have identified six critical elements to consider if a private health insurance model continues to be the preferred model. These critical elements are as follows:It should support and encourage participation in chronic disease prevention interventions including screening for hyperlipaemia, diabetes mellites, hypertension, and selected cancers as appropriate for different segments of the population. Studies have shown that using preventative health services can reduce morbidity and mortality for individuals and improve the risk profile of insured populations [44,45].It should maintain the principle of community rating to prevent insurers from discriminating against members based on age, health status, or claims history [46]. This is particularly important for NCDs, as a substantial proportion of the population will have existing chronic conditions at the time of entry to the insurance pool.It should assure that a similar level and quality of healthcare is available to all participants and that enrolment and coverage is not affected by pre-existing conditions, variable co-payments and deductibles, and mandatory preauthorization of costly investigations and procedures. Consideration should be given to how the system rewards an appropriate care provision while disincentivising over-servicing, a significant concern in any fee-for-service remuneration model.Funding mechanisms should fund and promote integrated care [46] to engage more effectively with primary, secondary, and/or tertiary prevention of NCDs. The ability of PHI to engage in NCDs prevention is increasing [47], not only in programmes focusing on primary and secondary prevention, but also in self-management, which plays a critical role in successfully treating chronic illnesses to prevent the recurrence of symptoms or consequences [48]. Saudi Arabia has developed a very successful high coverage primary care sector and it is important this is not lost in a private health insurance model. Good integration of services means that patients with NCDs can be appropriately and well managed in the relatively lower cost setting of primary care while retaining ease of access to higher level and more costly care when required.There should be an integrated comprehensive individual electronic health record that is accessible to clinicians and patients wherever it is needed in the health system, that is, across public and private providers. This will assist with continuity of care, assists monitoring quality of care, and helps to reduce unnecessary duplicative servicing.Insured services should include the full range of clinical care required by patients with NCDs, including allied health services, such as dietetics, as well as supporting patient engagement in self-care through patient education and community support programs.

### 4.3. Weaknesses

Respondents in this study reported that the major area for reform with the current funding model other than sustainability was the current healthcare provider payment. Provider payment methods in healthcare are critical and should reflect providers’ performance, safety, and quality of care delivered to patients [49,50]. Private health insurance plans in Saudi Arabia usually remunerate through the FFS form of payment. This payment mechanism is superficially straightforward and has certain benefits, such as encouraging the delivery of care and incentivising physicians to see more patients. However, the fee structure is unregulated, and incentivising servicing can lead to overprovision, inefficiency, and consequently high health costs [51]. Physicians may disregard simple and often equally effective treatments procedures because they are poorly reimbursed and time-consuming to perform.

A salary payment model is used by the vast majority of public hospitals to compensate their healthcare professionals [6,9]. While the salary payment model can better control the financial impacts of health professional costs, there are concerns that it may reduce individual professional productivity, as there is no incentive for healthcare professionals to see the optimal number of patients [52]. There are mixed salary–FFS models and other models, such as capitation, that should be considered with any move to change the funding model. The need for policymakers to consider these other options is supported both by respondents in our study and other studies [39,53].

In deciding on a new payment or remuneration model, the literature suggests that the following criteria are important: 1. Providing care that is accessible, safe, and effective; 2. Ensuring efficient healthcare delivery in order to control unnecessary utilisation; 3. Basing compensation on recognised quality of treatment; 4. Taking into account the cost, time, and complexity of treating each patient; 5. Reducing physician incentives to perform diagnostic tests that are both expensive and ineffective; and, 6. Promoting and increasing the accountability and appropriateness of care [54,55,56].

### 4.4. Challenges

This research suggests that there is a need to regulate the provision and mechanisms of healthcare services to improve cooperation and coordination between private and government care sectors to avoid duplication of access to healthcare services. Furthermore, the current care model that accepts patients without regulating the provision and mechanism of healthcare services results in duplication of access to healthcare services, which results in the inequitable distribution of healthcare resources. Examples include the fact that, on the one hand, some Saudis who work in private sectors are eligible to obtain private health insurance, but, on the other hand, they are also entitled to free healthcare services in public and semi-public healthcare facilities. In contrast, Saudis who work in government sectors are entitled to healthcare services solely at public healthcare facilities. This unregulated mechanism is inequitable to all individuals involved. As a result, if an individual is eligible for mandated health insurance, he or she may be able to receive healthcare services in public healthcare facilities without incurring any financial obligations on the insurance company. Therefore, individuals with co-payment insurance will utilise public healthcare facilities more often, while health insurance companies will be able to optimise and maximise profits.

## 5. Conclusions

The healthcare financing system in Saudi Arabia faces challenges common to many countries; increased costs due to demographic changes, inequity, an ageing population, an increase in NCDs, escalating costs of healthcare services, and increased public demand for improved healthcare are issues facing the public sector. There is concern that existing public revenue will not be able to support the increased demand under the existing largely government-funded approach. The proposed solution to this is to move to a largely private health insurance model, the details of which are unclear. From the perspective of NCDs, probably the largest determinant of healthcare demand, there are risks in such an approach in terms of both quality of care, care coverage, and cost drivers.

We have identified essential elements of NCD care that should be considered in moving to any new funding model, including private health insurance ensuring continuity of care and equity, funding chronic disease prevention interventions, prioritising primary healthcare, and maintaining the principle of community rating to prevent insurers from discriminating against members. Other desirable attributes include funding mechanisms that promote integrated care to engage effective and efficient collaboration across primary, secondary, and tertiary care. Without attention to these elements, there is a substantial risk of poorer outcomes for health, equity of health, and for finances from a move to a private health insurance model.

## Figures and Tables

**Table 1 healthcare-10-02294-t001:** Sociodemographic and medical characteristics of the 315 respondents.

Variable	Frequency	%
Age category		
Below 35	10	3.2
35–49	64	20.3
50–64	165	52.4
65 and above	76	24.1
Gender		
Male	157	49.8
Female	158	50.2
Level of education		
Not attended	34	10.8
Primary school	52	16.5
High school	77	24.4
Diploma	29	9.2
Graduate	117	37.1
Postgraduate	6	1.9
Occupation		
Government	115	36.6
Private	49	15.6
Unemployed	73	23.3
Retired	77	24.5
Monthly income		
<3000 SAR	60	19.1
3000–5000 SAR	44	14.1
5000–8000 SAR	84	26.8
>8000 SAR	126	40.0
Number of NCD’s having (*n* = 315)		
Single NCD	238	75.6
Having two or more NCD’s	77	24.4
(Among those having a single NCD) Disease (*n* = 238)		
Cardiovascular disease	39	16.4
Chronic respiratory disease	38	16.0
Diabetes	101	42.4
Hypertension	51	21.4
Stroke	5	2.1
Other	4	1.7
Duration of illness		
5 years or below	140	44.4
6–10 years	80	25.5
11–20 years	84	26.8
>20 years	10	3.3

**Table 2 healthcare-10-02294-t002:** Patients’ perception of private health insurance *n* = 315.

Independent Variable		Happy to Have Private Health Insurance to Access Any Hospital *p*-Value
Age		<0.01 *
Gender (Reference—Male)	Female	<0.01 *
Education (Reference—Postgraduate)	Not attended	0.99
	Primary school	0.99
	High school	0.99
	Diploma	0.99
	Graduate	0.99
Monthly income (Reference—>SAR 8000)	<3000	0.43
	3000–5000	0.20
	5000–8000	0.87
Occupation (Reference—unemployed)	Private	0.08
	Government.	<0.01 *
Duration of Illness (Reference—>20 years)	<5 years	<0.02 *
	6–10 years	<0.02 *
	11–20 years	<0.01 *

* *p* < 0.05.

## Data Availability

The data presented in this study are available on request from the corresponding author.

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
