# Peer review of "Reform of the Health Insurance Funding Model to Improve the Care of Noncommunicable Diseases Patients in Saudi Arabia"

_healthcare, 2022, doi:10.3390/healthcare10112294_

Round 1

Reviewer 1 Report

Comments on a manuscript entitled “Reform of the health insurance funding model to improve the care of noncommunicable diseases patients in Saudi Arabia

General comments

This study assessed the attributes of health funding models to support non-communicable diseases (NCDs) control in Saudi Arabia. Some issues should be addressed before acceptance for publication.

Major comments:

-          The authors might need to provide semi-structure interview and survey questionnaires validation. How did the authors validate the forms?

-          Were the forms valid and reliable? The authors might need to provide statistics to ensure validity and reliability of both interview and survey forms.

-          Which diseases were defined as NCD? The authors might need to provide details on the NCDs included in this study.

-          Details on the inclusion criteria for the survey should be provided. What age of patients? Were there any exclusion criteria?

-          How many PHCs were included in the survey?

-          The details on the survey forms should be briefly explained.

-          Sample size calculation should be provided to ensure the representativeness of the sample.

-          The details of multiple logistic regression should be provided. How did the authors select covariates; systematic review? How did the authors select covariates for the final model?

-          Tests used for p-value of Table 1 should be provided.

-          Table 1 should provide crude data of each level of each variable with p-value. Only p-value might not be enough to show the perception of patients.

-          Was the interview face-to-face or groups or online?

-          The authors might need to provide blank survey and interview forms as supplement.

Author Response

We greatly appreciate the reviewers’ enthusiasm about manuscript and the quality comments offered by the reviewers. The suggestions offered by the reviewers have been immensely helpful, and we also appreciate your insightful comments on revising the paper. 

Major comments:

-          The authors might need to provide semi-structure interview and survey questionnaires validation. How did the authors validate the forms?

Noted and changes incorporated in the manuscript.

The semi-structured interview guide was developed after reviewing the relevant literature on NCDs and healthcare systems in Saudi Arabia. The participants’ interview guide included questions regarding their views with current financing system for NCDs and the role of healthcare insurers in the financing of NCDs care. The semi-structured interviews were pilot tested and reviewed for content validation by up to nine experts working at the NCDs prevention and management departments. The patient assessment of chronic illness care (PACIC) was used to assess the quality of care received by patient with NCDs. PACIC was selected for this study because it allows for a standardised assessment of the patient’s perspective on the current quality of health care services, which is essential in the evaluation of chronic care management [22,23]. The PACIC questionnaire has been validated and used in the Saudi context [24].

-          Were the forms valid and reliable? The authors might need to provide statistics to ensure validity and reliability of both interview and survey forms.

Note and changes made in the method part.

-          Which diseases were defined as NCD? The authors might need to provide details on the NCDs included in this study.

Details on the NCDs were provided in table 1. And briefly described as well.

The final sample consisted of 315 patients with NCDs. In brief, the mean age of the patients was 56years, ranging from 29 to 85; 49.8% were male and 50.2% were female, with more than half (52.4%) belonging to the 50 to 64 age group. Graduate level education was reported by over a third (37.1%), and 44.4% reported having been diagnosed with an NCD for less than 5years. Among the study sample, 75.6% reported having only 1 NCD, while 24.4% had more than 1 NCD. Diabetes was the most common NCD, reported by 43% of the study population, followed by hypertension in 21.4% patients. Almost 50% of the diabetes patients also had hypertension. The presence of NCDs among the patients is given in Table 1.

-          Details on the inclusion criteria for the survey should be provided. What age of patients? Were there any exclusion criteria?

The ages of the patients were provided in the Table 1, also the inclusion criteria has been included.

Only patients with NCDs were recruited for inclusion in the study. The inclusion criteria were patients with at least one chronic disease and who had attended a NCDs clinic at a PHC during the previous six months. All patients meeting these criteria and attending a clinic on study days were approached to participate in the study.

-          How many PHCs were included in the survey?

Noted and changes made,

The patients were recruited from 20 of 58 Ministry of Health (MOH) PHCs in urban Riyadh, Saudi Arabia.

-          The details on the survey forms should be briefly explained.

Noted and incorporated in the manuscript.

The semi-structured interview guide was developed after reviewing the relevant literature on NCDs and healthcare systems in Saudi Arabia. The participants’ interview guide included questions regarding their views with current financing system for NCDs and the role of healthcare insurers in the financing of NCDs care. The semi-structured interviews were pilot tested and reviewed for content validation by up to nine experts working at the NCDs prevention and management departments. The patient assessment of chronic illness care (PACIC) was used to assess the quality of care received by patient with NCDs. PACIC was selected for this study because it allows for a standardised assessment of the patient’s perspective on the current quality of health care services, which is essential in the evaluation of chronic care management [22,23]. The PACIC questionnaire has been validated and used in the Saudi context [24].

-          Sample size calculation should be provided to ensure the representativeness of the sample.

Noted and incorporated in the manuscript.

For an estimated patient population of 20 000 the sample size for estimates of PACIC scores with a 5% margin of error at 95% confidence level, was 377 (calculated using Raosoft calculator)[25] simplified as a target of 400. The sample size achieved was 315 due to incomplete questionnaires giving an effective response rate of 79%. This sample size is similar to that used to develop and validate the PACIC instrument [26].

-          The details of multiple logistic regression should be provided. How did the authors select covariates; systematic review? How did the authors select covariates for the final model?

Based on the characteristics of the participants in the study which reported in the table 1

-          Tests used for p-value of Table 1 should be provided.

Binary logistic regression was performed for the association between socio-demographic and disease related variables with selected aspects of patient experience.

-          Table 1 should provide crude data of each level of each variable with p-value. Only p-value might not be enough to show the perception of patients.

The crude data will be provided as supplementary.

-          Was the interview face-to-face or groups or online?

Face-to-face and this has been incorporated in the manuscript.

A sample of 25 health providers and managers from the  MOH were interviewed face-to-face at their workplaces.

-          The authors might need to provide blank survey and interview forms as supplement.

The blank survey and interview forms  will be provided as supplementary

Reviewer 2 Report

The abstract is laconic. The abstract should be changed and written according to the accepted standards: research objective, research methods and tools, research problem and hypothesis, and the most important conclusion from the research (the abstract should encourage reading and, consequently, publishing the article).

The aim of the research, the problem and the research hypothesis should be specifically articulated, also in the introduction.

There are no specific conclusions resulting from the conducted research.

The article is more descriptive. There is no statistical analysis which would confirm the nature of the study.

Perhaps the existing health care system in Saudi Arabia should be compared with another country, similar to the system described by the Authors.

It is limiting whether the readers will be interested in the problem of the health care system and Saudi Arabia - I leave it to the editorial decision.

Improve the language.

Author Response

We greatly appreciate the reviewers’ enthusiasm about manuscript and the quality comments offered by the reviewers. The suggestions offered by the reviewers have been immensely helpful, and we also appreciate your insightful comments on revising the paper. We have included the reviewers’ comments in the following table and responded to them individually, describing the changes we have made

The abstract is laconic. The abstract should be changed and written according to the accepted standards: research objective, research methods and tools, research problem and hypothesis, and the most important conclusion from the research (the abstract should encourage reading and, consequently, publishing the article).

The abstract has been formed according to the journal guideline and include the key findings of the study.

The aim of the research, the problem and the research hypothesis should be specifically articulated, also in the introduction.

Noted, kindly see the last paragraph of the introduction.

This paper aims to assess the desired attributes for any health financing model to improve the control and management of NCDs. It assesses the perspective of patients with NCDs, clinicians and policymakers on the reform of the Saudi health insurance funding model. Given the planning and decision making for the Saudi funding model is ongoing, this study aims to inform decisions about the design of any health insurance model to improve and not be a barrier to a more effective and efficient health system approach to NCDs.

There are no specific conclusions resulting from the conducted research.

In conclusion, the article identified essential elements of NCDs care that should be considered in moving to any new funding model including private health insurance including ensuring continuity of care and equity, funding chronic disease prevention interventions, prioritising primary health care, and maintaining the principle of community rating to prevent insurers from discriminating against members. Other desirable attributes include funding mechanisms that promote integrated care to engage effective and efficient collaboration across primary, secondary, and tertiary care. Without attention to these elements there is a substantial risk of poorer outcomes for health, equity of health and for finances from a move to a private health insurance model.

The article is more descriptive. There is no statistical analysis which would confirm the nature of the study.

A binary logistic regression analysis was performed for the association between socio-demographic and disease related variables with selected aspects of patient experience. Statistical significance of the results was accepted with a P-value ≤ 0.05 as shown in Table 2.

Perhaps the existing health care system in Saudi Arabia should be compared with another country, similar to the system described by the Authors.

The study reports on the best practices and procedures based on high-quality evidence to obtain improved patient and health outcomes when implementing Health Insurance. The study identified six critical elements to consider if a private health insurance model continues to be the preferred model for any healthcare system.

1) It should support and encourage participation in chronic disease prevention interventions including screening for hyperlipaemia, diabetes mellites, hypertension and selected cancer as appropriate for different segments of the population. Studies have shown that using preventative health services can reduce morbidity and mortality for individuals and improve the risk profile of insured populations [44,45].

2)  It should maintain the principle of community rating to prevent insurers from discriminating against members based on age, health status or claims history [46]. This is particularly important for NCDs as a substantial proportion of the population will have existing chronic conditions at the time of entry to the insurance pool.

3) It should assure similar level and quality of health care is available to all participants and that enrolment and coverage is not affected by pre-existing conditions, variable co-payments and deductibles, and mandatory preauthorization of costly investigations and procedures. Consideration should be given to how the system rewards appropriate care provision while disincentivising over-servicing, a significant concern in any fee-for-service remuneration model.

4) Funding mechanisms should fund and promote integrated care[46], to engage more effectively with primary, secondary and/or tertiary prevention of NCDs. The ability of PHI to engage in NCDs prevention is increasing [47] , not only in programmes focusing on primary and secondary prevention, but also in self-management, which plays a critical role in successfully treating chronic illnesses to prevent the recurrence of symptoms or consequences [48].  Saudi Arabia has developed a very successful high coverage primary care sector and it is important this is not lost in a private health insurance model. Good integration of services means that patients with NCDs can be appropriately and well managed in the relatively lower cost setting of primary care while retaining ease of access to higher level and more costly care when required. 

5) There should be an integrated comprehensive individual electronic health record that is accessible to clinicians and patients wherever it is needed in the health system, that is, across public and private providers. This will assist with continuity of care, assists monitoring quality of care, and helps reduce unnecessary duplicative servicing.

6) Insured services should include the full range of clinical care required by patients with NCDs including allied health services such as dietetics, as well as supporting patient engagement in self-care through patient education and community support programs.

It is limiting whether the readers will be interested in the problem of the health care system and Saudi Arabia - I leave it to the editorial decision.

The study findings report on the best practices and procedures based on high-quality evidence to obtain improved patient and health outcomes when implementing Health Insurance. It benefits other healthcare systems by highlighting the main elements to be considered in any private health insurance Model.

Improve the language.

The manuscript has been critically analysed and reviewed.  

Round 2

Reviewer 2 Report

I leave the decision to publish the article to the editors, because, as I pointed out in the review: it is not very original and may be of interest to the readers